# Wear and Service Life of 3-D Printed Polymeric Gears

**DOI:** 10.3390/polym14102064

**Published:** 2022-05-18

**Authors:** Mert Safak Tunalioglu, Bekir Volkan Agca

**Affiliations:** 1Deptartment of Mechanical Engineering, Faculty of Engineering, Hitit University, 19100 Çorum, Turkey; 2Institute of Science, Hitit University, 19030 Çorum, Turkey; bvolkanagca@gmail.com

**Keywords:** plastic spur gear, wear, FZG, service life, ABS, PLA, PETG

## Abstract

Plastic gears are mostly used in the textile, food, and automotive industries due to their silent operation, corrosion resistance, and light and cheap advantages. Plastic gears are generally manufactured by injection molding or hobbing methods. The excess costs of the molds used to produce parts in injection molding and the problems of wastes that occur during production in hobbing lead companies to additive manufacturing, which is an alternative application. In the additive manufacturing method, the desired amount of product is produced without the problem of waste. In this study, the wear resistance of plastic spur gears produced by the Fused Deposition Modeling (FDM) method was determined theoretically. In order to determine the service life of gears, wear tests were carried out in the Forschungsstelle fur Zahnrader und Getriebebau (FZG) type test device at the same load and rotational speeds. polylactic acid (PLA), acrylonitrile butadiene styrene (ABS), and polyethylene terephthalate (PETG) thermoplastic polymer materials were used in the production of gears. When the gears rotate at the same load and rotational speeds, the most wear was observed in ABS, PLA, and PETG at the theoretically calculated wear depths. PETG is the most resistant material in terms of wear.

## 1. Introduction

The additive manufacturing method has been preferred more than traditional production methods in recent years. One of the most important reasons for this is that raw materials are used just as much as the product to be produced in the additive manufacturing method; thus, there is no need to eliminate waste materials that occur during production. While more than one process and machine are needed to produce a product in traditional manufacturing methods, the product is easily produced with a single process and machine in the additive manufacturing method.

Plastic materials are generally produced by hot isostatic pressing (HIP), sintering (PS), and injection molding (IM) methods. In addition to the advantages of producing many parts quickly, there are disadvantages such as the cost of the mold due to the need for a different mold for each part and the need for stock space for the large number of parts produced. In the additive manufacturing method, the desired number of parts is produced without the need for molding. The most important disadvantage of the additive manufacturing method is that the production time is longer than the traditional method and it is not suitable for mass production. However, satisfactory studies have been carried out on this subject in recent years.

It is possible to produce products that have been modeled with a three-dimensional printer with a low investment cost. The operation of the system is to produce products by pouring the molten polymeric materials in layers. In this system, PLA, ABS, PETG and various thermoplastic materials can be used.

Plastic gears are frequently used especially in the food, electrical, and textile industries due to their lightness, corrosion resistance, cheapness, and quiet operation. For example, cutters in the textile and food industry, covers and gears in electric motors, and even pulp dentin, implants, and some tissues with 3D bioprinting can be easily fabricated with 3D printers [1]. It is important to know the wear resistance and service life of such materials, as they are not resistant to extreme loads and temperatures.

Researchers have compared the mechanical properties of plastic materials produced in three-dimensional printers [2,3,4,5,6,7,8,9,10,11,12,13,14,15,16,17,18,19]. Vanaei et al. [6] studied the interactions of materials produced with three-dimensional printers with temperature. Amirruddin et al. [7] investigated the tribological properties of 3D printed pins manufactured at different raster angles and layer thicknesses. FDM printed acrylonitrile butadiene styrene (ABS) and Polylactic Acid (PLA) samples were printed at three raster angles of 0°, 45°, and 90° and in three layer thicknesses of 0.127 mm, 0.254 mm, and 0.33 mm. The experiments were carried out on a pin-on-disc tribometer apparatus under dry sliding condition and constant load of 10 N and 300 rpm. The results showed that increasing the layer thickness favored a reduction in friction force, while an angle of 45° showed the best wear resistance. ABS has better wear resistance than PLA. To examine the friction and wear behavior of the plastic materials produced by the FDM method, wear tests were carried out on the Pin on Disc assembly [9,10,11,12]. Some researchers compared PLA with ABS [15,16,17], and ABS with PETG [18,19] in the form of production stages and wear resistance. Roy et al. [17] investigated the friction and wear behavior of ABS and PLA thermoplastics with the FDM method. Material deposition layer thickness, infill angle, infill pattern and deposition direction, and main printing parameters were used in their studies. As a result of their experiments the service life of PLA parts increasing at a high rate.

Researchers have studied the friction, wear, and service life of gears produced by plastic injection molding [20,21,22,23,24,25,26,27,28,29,30,31]. Singh et al. [23] produced ABS, HDPE, and POM gears by the injection molding method and their thermal and wear behaviors were investigated. While ABS and HDPE gears completed 0.5 and 1.1 million cycles, respectively, before failure, they claimed that the POM gear completed 2 million cycles without any signs of failure. Duzcukoglu and Imrek [24,25] increased the tooth width in PA 66 gears, delaying the occurrence of thermal damage in the single tooth region. Experiments show that the appearance of heat damage is delayed for width modified gear teeth compared with unmodified gear teeth. Kalin et al. [27] investigated the tribological properties of POM gears at different temperatures and torques. Tsai et al. [28] calculated the static transmission errors of plastic gears using the FEM method. Theoretical and experimental research by Senthilvelan et al. [30] were carried out to investigate the surface smoothness and part shrinkage of non-reinforced and glass fiber reinforced Nylon 6/6 gears produced by plastic injection molding. Dhanasekaran et al. [31,32] investigated the wear properties of sintered gears containing molybdenum disulfide powders. They stated that the compositions containing MoS_2_ increased the tooth strength and wear resistance. Letzelter et al. [33] developed a program to predict the mechanical behavior of polyamide 6.6 gears. In their studies, the viscoelastic behavior of gears was investigated theoretically and experimentally.

Researchers have examined the strength of plastic gears produced in three-dimensional printers [34,35,36,37,38,39,40,41,42,43,44]. The mechanical properties of gears produced with the 3D printer technique have been studied theoretically and experimentally under different operating conditions [34,35]. Zhang et al. [36] produced spur gears with Nylon, Nylon 618, Nylon 645, and alloy 910 filaments in three-dimensional printers. They stated that Nylon 618 material is the most durable. In addition, they stated that Nylon 618 material outperformed Nylon 66 material manufactured with plastic injection at low to medium torque values. Aleksandar et al. [37] produced spur gears from PLA and ABS materials with the help of 3D printer technique and performed wear tests at different speeds. As a result of their experiments, they stated that PLA material is more durable than ABS material. Dennin et al. [40] compared the production, quality, and wear tests of gears fabricated of PLA, ABS, PA, and PA carbon fiber composite materials with the FDM method. At the end of their study, PLA gears failed due to low melting temperature, while ABS and PA gears failed in root fracture. Harsha et al. [42] tested PLA, ABS, and Nylon gears produced on three-dimensional printers. In the tests, they found the wear coefficient at 600, 800, and 1100 rpm, and 1.3 Nm loads. As a result of their tests, they found that the PLA gears broke at 1100 rpm. In the comparison of all three polymeric gears, they stated that the most durable gear was the Nylon gear. Feteke [43] investigated the friction coefficients of PLA gears at different temperatures. As a result, he said, replacing the CoF can reduce wear by 35% in this specific area, while even a 5 °C increase in temperature results in 40% wear progression. Tunalioglu et al. [44] performed some tests to find wear along the line of action in PLA gears. 

In this study, the wear of the plastic gears produced by the FDM method in three-dimensional printers was determined theoretically according to the number of rotations. The equation determined to describe the wear along the line of action is solved with the help of MATLAB^®^. The service life of the gears was determined by continuing the tests at the same load and rotational speeds.

## 2. Materials and Methods

### 2.1. Wear Model in Gears

Gears in contact with each other make sliding and rolling movements. As a result of these movements, wear occurs on the surfaces of the gears. Wear shows different features at different points depending on the contact ratio on the teeth.

Wear on a tooth surface varies depending on the sliding speeds from the dedendum to the addendum and the distribution of the load on the gear pair. According to Figure 1a, point A is the root of the tooth where the gear enters the clutch, point E is the addendum region where the gear leaves the clutch. Points D and B are the single tooth contact areas and P point is the pitch point where the speeds are equal and where wear is not observed only due to rolling. Gears make double tooth contact in E-D and A-B regions depending on the contact ratio. Therefore, in these regions the load is shared by the two teeth and wear is less in these regions (Figure 1b).

Wear due to load on objects has been described by Archard [45]. According to the Archard wear equation:(1)$$\frac{V}{s}=K\frac{W}{H}$$

In this equation, (*V*) is volume of the abraded material, (*s*) is slippage distance between two surfaces, (*K*) is the friction coefficient, (*W*) is applied load, and (*H*) is the surface hardness of the material. Flodin described the Archard wear equation from the beginning to the end of the contact in the gear pair [46]. According to this equation:(2)$$h_{p}=\int_{0}^{s}kPds$$
where (*h_p_*) is the wear depth in point (*p*), (*s*) is the sliding distance between contacting surfaces, (*k*) is the wear coefficient, and (*P*) is the contact pressure. If this equation is calculated according to a certain rotational speed along the contact length of any point (p) in contact:(3)$$h_{p,n}=h_{p,\left(n-1\right)}+kP_{p,\left(n-1\right)}s_{p}$$
where (*h_p,(n − 1)_*) is the wear depth of point (*p*) in the previous cycle, (*k*) is the wear coefficient, (*P_p,(n − 1)_*) is the pressure in the previous cycle, and (*s_p_*) is the sliding distance of point (*p*). In this equation, the movements of points such as p_1_ and p_2_ taken on both gears in contact are important in order to find the slip velocity. There are three different situations here. The first situation is when both gears engage contact simultaneously. There is no slipping distance between points at this location. In the second case, p_1_ disengages the contact and there is a sliding distance between p_1_ and p_2_ (*S_p1_*). In the third case, p_2_ disengages the contact and there is a sliding distance between p_1_ and p_2_ (*S_p2_*). The slip distance occurring in the second and third cases is proportional to the speed of the gears (Figure 2).
(4)$$S_{p1}=2a_{H}\left(\frac{U_{1}-U_{2}}{U_{1}}\right)$$
(5)$$S_{p2}=2a_{H}\left(\frac{U_{2}-U_{1}}{U_{2}}\right)$$
where *U_1_* and *U_2_* are peripheral speeds of gears. If Equations (4) and (5) are written in Equation (3), wear depth equations for pinion and gear are, respectively:(6)$$h_{p,n}=h_{p,\left(n-1\right)}+kP_{p,\left(n-1\right)}2_{a_{H}}\left(1-\frac{U_{1}}{U_{2}}\right)$$
(7)$$h_{p,n}=h_{p,\left(n-1\right)}+kP_{p,\left(n-1\right)}2_{a_{H}}\left(\frac{U_{1}}{U_{2}}-1\right)$$

This equation expresses the wear of each point on the gears along the contact length depending on the number of rotations.

### 2.2. Materials and Test Procedure

In this study, PLA, ABS, and PETG polymeric thermoplastic materials, which are frequently used in the FDM method, were used. The mechanical properties of the materials are shown in Table 1.

PLA is obtained from materials such as corn and sugarcane. It has easy production in three-dimensional printers, low strength, and is brittle. ABS is a petroleum-based material. Its production is difficult compared with PLA. Its strength is higher when compared with PLA. PETG is higher than PLA and ABS in terms of strength. It is more difficult to manufacture than PLA. The production stages of polymeric materials produced with a three-dimensional printer and the gears produced are shown in Table 2 and Figure 3.

In the study, a BCN 3D Sigma R19 type three-dimensional printer was used to produce plastic gears (Figure 4). No support material was used in the printing process. The printing angle was taken as 0°. Since no support material was used during the production of the plastic gears, surface finishing processes were not carried out at the end of the production. Accuracy of the produced gears are less then ±0.05 mm. 

To observe the wear on the gears, wear tests were carried out with the FZG closed circuit wear test device (Figure 5). In the tests, a 1.5 Nm load was applied to the gear pair and the system was rotated at 900 rpm. The gears were rotated for a total of 10^5^ rotations to determine the theoretical wear. After every 10^4^ revolutions of the pinion gear, the gears were removed, cleaned, and weighed. Wear amount weighing of 10^−4^ gr. was performed with precision scales. At the end of the tests, the total wear amount was determined and the wear coefficient to be used in the theoretical equation for each gear was determined (*k*). To determine the service life of the test gears, they were run at the same load and rotational speeds until the gears were damaged. In order to observe the wear in polymeric gears, a St 37-2 steel gear was used as the driven gear. The characteristics of the gears used in the tests are given in Table 3.

In the tests an St37-2 steel gear was chosen as the driven gear. For plastic gears that work for a long time, wear occurs with applied load and sudden temperature rising. Damage to any tooth during action in gears working together quickly affects the whole system and plastic gears become inoperable. In addition, the wear coefficient in plastic gears is calculated by the weight loss of gears in the desired number of rotations. In cooperating plastic gears, the wear particles may not be completely separated from the environment with the temperature and may adhere to the driven gear. For these reasons, it is more appropriate to choose a steel gear for the driven gear when performing the wear tests on plastic gears.

## 3. Results and Discussion

### 3.1. Theoretical Wear of Polymeric Gears

To determine theoretical wear of PLA, ABS, and PETG polymeric spur gears produced by the FDM method in three-dimensional printers, the gears were subjected to wear tests. The tests continued until the pinion gear turned 10^5^ cycles. The wear depth values are shown in Figure 6.

According to Figure 6, the most wear in all three gears is seen in the root region where the plastic gears enter the mesh. Since the differences in the sliding speeds from the root of the tooth to the pitch circle become closer to each other, the wear decreases. Wear shows a sudden increase as the contact ratio acts from double to single tooth. No wear is observed in the pitch point as the gears only make a rolling motion. From the pitch circle to the exiting mesh the wear increases again gradually. The wear depths of the materials used in this study of plastic gears are compared, the most wear is observed as ABS, PLA, and PETG, respectively. The wear depths of ABS and PLA materials were close to each other. The wear depths of PETG material gears were 30% and 40% less than the ABS and PLA material, respectively. As a result, PETG is the most resistant material in terms of wear.

In the previous study [44], the wear depths of spur gears fabricated of PLA material by the FDM method were measured with the help of Coordinate Measuring Machines (CMMs) along the line of action. In this study, the wear depths along the action of the gears produced by the same method were determined theoretically with the help of the MATLAB^®^ program. When the two methods are compared for the same load and rotational cycles, the experimentally wear values are 4–5% higher on average. Considering the construction times of the tests, the removal of the gears from the device after each test stage and the measurements taken, and the excess of factors affecting the wear on the gears, the use of the theoretical equation is appropriate with this difference.

### 3.2. Service Life of Polymeric Gears

In order to determine the service life of the polymeric gears produced by the FDM method, the gears were operated at 900 rpm and 1.5 Nm load until damage occurred. The damage times and shapes of the gears are shown in Figure 7, Figure 8 and Figure 9.

Plastic gears are not resistant to high loads and temperature rises; it is very important to know or at least estimate their service life. In this study, low load was applied to plastic gears in life tests. The reason for this is that excessive wear was observed in the gears due to sudden temperature rises at high loads. As a result of the tests, the service life of the ABS and PLA materials were close to each other (Figure 7, Figure 8 and Figure 9). Plastic gears produced with PETG material are more durable than other materials. PETG plastic gears are 22.3% and 37% more durable than PLA and ABS, respectively. PLA plastic gears seem to be 12% more durable than ABS. At the end of the tests, the earliest damage was seen in the plastic gears fabricated of ABS material. It is thought that ABS material is harder and more brittle than other materials. It was observed that the most durable material was PETG. The most important reason for this is that the elongation at the break rate was 4–5 times higher than PLA and ABS; this shows that PETG material is more ductile and tougher than other materials. PLA and ABS materials showed similar wear behaviors.

## 4. Conclusions

In this study, the wear depth equation of the plastic gears produced by the FDM method in the three-dimensional printer was determined theoretically and the wear depths along the line of action were determined with the derived MATLAB^®^ code. To determine the service life of the gears, the tests were repeated depending on the same speed and number of revolutions until damage occurred.

In the plastic gears produced by the FDM method, the wear was mostly observed in the tooth root region where the driving gear entered action. The critical point for plastic gears is the root region of the gears. The wear depth values calculated as a result of the theoretical equation for gears fabricated of PLA material differed by 4–5% with the results measured experimentally with the help of CMM. This shows the usability of the theoretical equation and its applicability in other plastic gears. It was observed that ABS and PLA exhibit similar wear behaviors. When all three plastic gear materials are compared, it can be said that the most durable material is PETG. The fact that PETG material is resistant to wear is due to its high elongation break and because the material is hard but flexible.

Three different polymeric materials (PLA, ABS, and PETG) most used in the production of materials with the FDM method in three-dimensional printers were compared due to their different properties. While the simplest production is fabricated with PLA material, ABS material is relatively more difficult to produce. When comparing price and availability, all three polymeric materials are usually sold as filaments and their costs are approximately the same. PETG is the most preferred material when wear resistance is compared.

In this study, the effect of material properties of three different polymeric materials on wear was investigated theoretically. In future studies, the wear behavior of polymeric gears produced with different three-dimensional printing techniques (SLA, SLS, POM etc.) and printing parameters (infill densities, tooth geometries, and raster angles) can be examined comparatively.

## Figures and Tables

**Figure 1 polymers-14-02064-f001:**
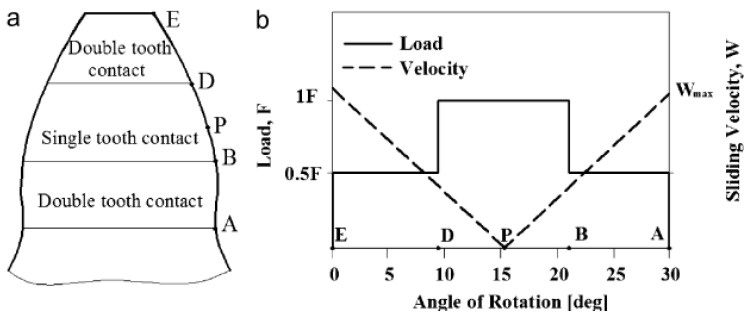
(**a**) Tooth contact areas and limits (**b**) load-surface velocity distribution [25].

**Figure 2 polymers-14-02064-f002:**
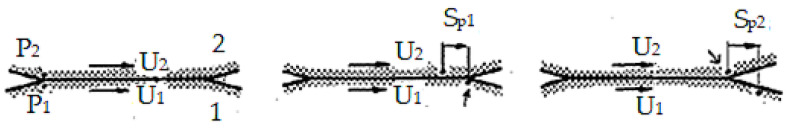
The relative movements of point p_1_ and p_2_ [47].

**Figure 3 polymers-14-02064-f003:**
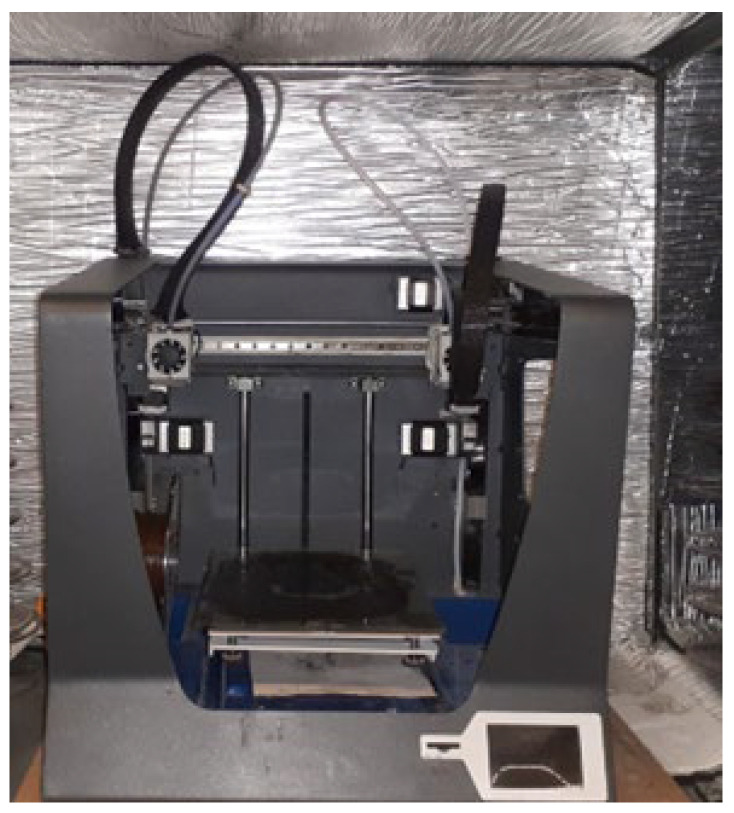
FDM type 3D Printer.

**Figure 4 polymers-14-02064-f004:**
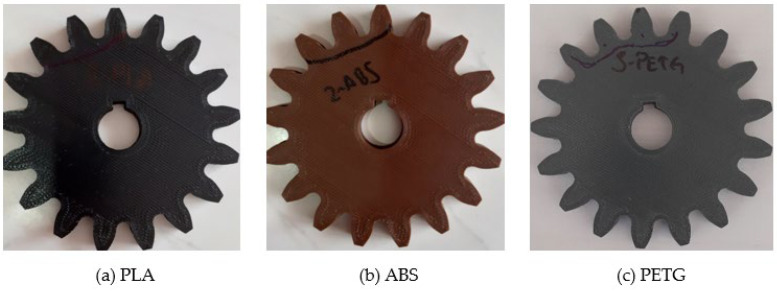
Polymeric spur gears (**a**) PLA, (**b**) ABS, (**c**) PETG.

**Figure 5 polymers-14-02064-f005:**
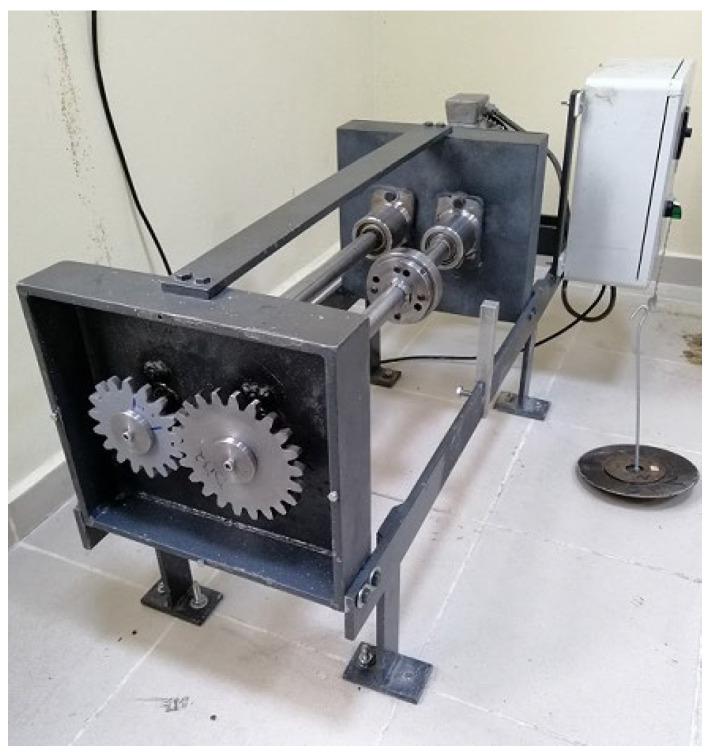
Wear test rig.

**Figure 6 polymers-14-02064-f006:**
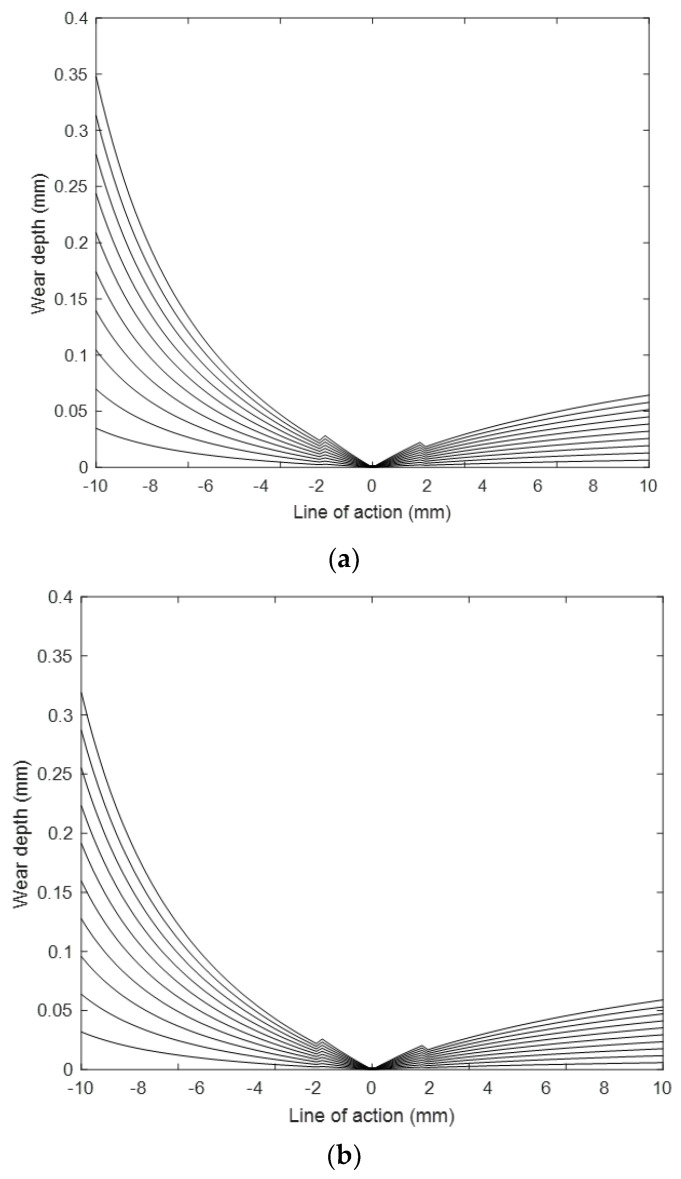
Wear depth variations through tooth profile of plastic gears (**a**) ABS, (**b**) PLA, and (**c**) PETG.

**Figure 7 polymers-14-02064-f007:**
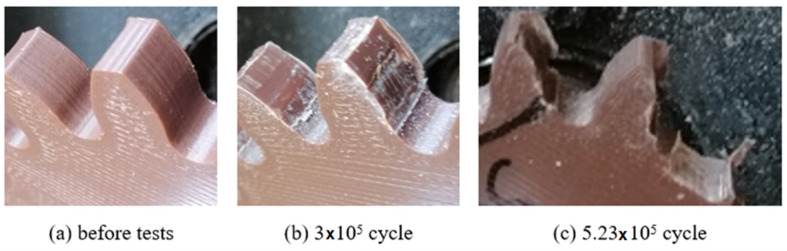
Service life of gears fabricated of ABS.

**Figure 8 polymers-14-02064-f008:**
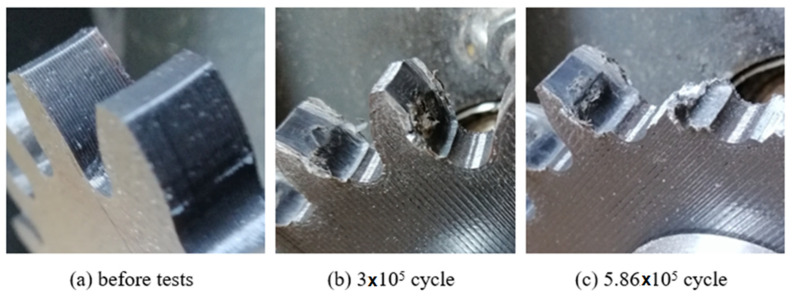
Service life of gears fabricated of PLA.

**Figure 9 polymers-14-02064-f009:**
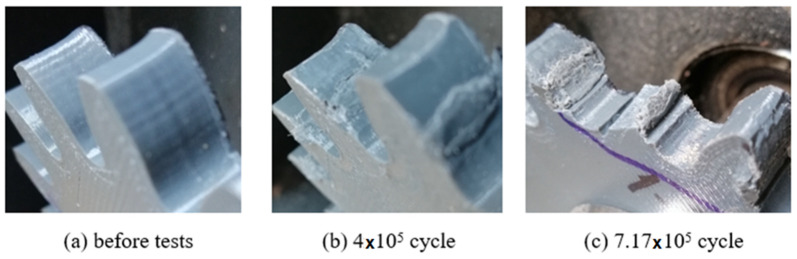
Service life of gears fabricated of PETG.

**Table 1 polymers-14-02064-t001:** Properties of polymeric materials.

Material and Test Method	Test Method	PLA	ABS	PETG
Density (g/cm^3^)	-	1.2	1.06	1.27
Tensile Strength (MPa)	ISO 527	62	40	50
Hardness (Shore D)	ISO 868	71	82	79.2
Poisson’s Ratio	-	0.35	0.35	0.43
Elongation at Break (%)	ISO 527	21.8	30	110
Shear Modulus (GPa)	ISO 527	2.4	0.88	4.6
Heat Deflection Temperature (°C)	ISO 75	53	73	70
Heat Deflection Temperature (J/kg-K)	ISO 75	1800	2000	1500
Thermal Conductivity (W/m-K)	ASTM C1045	0.13	0.17	0.21
Glass Transition Temperature (°C)	ISO 11357	60	105	80

**Table 2 polymers-14-02064-t002:** Production stages of plastic gears.

Material	PLA	ABS	PETG
Nozzle diameter (mm)	0.4	0.4	0.4
Filament diameter (mm)	2.85	2.85	2.85
Layer height (mm)	0.2	0.2	0.2
First layer height (mm)	0,19	0.19	0.19
Shells (mm)	4	4	4
Infill density (%)	100	100	100
Fill pattern	Rectilinear	Rectilinear	Rectilinear
Extruder temperature (°C)	210	245	246
Bed temperature (°C)	60	85	70
Printing speed (mm/s)	60	60.1	60.1

**Table 3 polymers-14-02064-t003:** Test gears.

Materials	PLA-ABS-PETG	St37-2
	Pinion	Gear
Modulus of elasticity (MPa)	3500	210,000
Module	6	
Number of teeth	17	22
Pressure angle (deg.)	20	
Profile shift correction	0	
Dia. of pitch circle (mm)	102	132
Dia. of tip circle (mm)	114	144
Tooth width (mm)	10	

## Data Availability

Not applicable.

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
