# Peer review of "Wear and Service Life of 3-D Printed Polymeric Gears"

_polymers, 2022, doi:10.3390/polym14102064_

Round 1
Reviewer 1 Report
Review reportManuscript ID: polymers-1710028
Manuscript title: Wear and Service Life of 3-D Printed Polymeric Gears
In this study, authors have determined theoretically the wear resistance of plastic spur gears using 3D
printing process. The paper and the details are interesting and I propose this paper for publication after
addressing the following issues and suggestions:
1. Introduction: In the first paragraph, I propose to explain in 2-3 lines the different application of AM
in industries such as automotive, biomedical, aviation, and so on. You can consider the following
reference for application in biomedical part:
https://doi.org/10.3390/polym13244442
Also, I propose to add a brief explanation regarding the influence of different process parameters that
exist in 3D printing process. It is useful to include it before entering the literature review through the
main subjects related to your study. Please consider the following reference as the importance of
temperature and the different sources that exist in this process:
https://doi.org/10.1016/j.jmapro.2022.02.042
2. Figure 2: is it possible to enhance the quality of the figures ? They are not in a god quality.
3. Figure 5, 6, 7: I think it is better to consider them as one figure and to consider them as subfigures as
a, b, c.
The paper is interesting and I propose it for publication. It has been well organized and discussed. The
findings are well classified and there is not too much comments to be considered for revision.
Best wishes in your work
Author Response
Point 1: Introduction: In the first paragraph, I propose to explain in 2-3 lines the different application of AM in industries such as automotive, biomedical, aviation, and so on. You can consider the following reference for application in biomedical part:
https://doi.org/10.3390/polym13244442
Also, I propose to add a brief explanation regarding the influence of different process parameters that
exist in 3D printing process. It is useful to include it before entering the literature review through the
main subjects related to your study. Please consider the following reference as the importance of
temperature and the different sources that exist in this process:
https://doi.org/10.1016/j.jmapro.2022.02.042
Response 1: Thank you for your comment. This paragraph added in Line 47-50. ‘For example, cutters in the textile and food industry, covers and gears in electric motors, and even pulp dentin, implants and some tissues with 3D bioprinting can be easily made with 3D printers [1].’
One of the reviewers suggested that the Introduction and Literature section was too long and we should keep it short. Thank you for your comment. We mentioned the parts you wanted in the first paragraph of the Literature section. We used the article you suggested in that section in Line 53-54. ‘Vanaei et al. [6] studied the interactions of materials produced with three-dimensional printers with temperature.’
Point 2: Figure 2: is it possible to enhance the quality of the figures ? They are not in a god quality.
Response 2: Thank you for your comment. We are sorry that we can't change that figure because we got it from the original article. But we found a clear one and change it.
Point 3: Figure 5, 6, 7: I think it is better to consider them as one figure and to consider them as subfigures as a, b, c.
Response 3: Thank you for your comment. Figures 5, 6, and 7 were rewritten as a single figure (Line 234). a, b and c were added as subtitles to the figures (Line 216).

Reviewer 2 Report
The article entitled “Wear and Service Life of 3-D Printed Polymeric Gears” by Tunalioglu et. Al. is well conceptualised and well written article. However there are few clarifications and changes that are necessary.
Line 181: please mention the model of 3D printer used. Also mention if any support material used, also mention the printing angle of the gear with respect to the print bed
Table 1 : please mention the standards used for measuring the properties (tensile strength etc).
Line 193 : Evaluation of 3D printed gears in terms of accuracy and surface finish would be appreciated.
Figure 1 : the image is blur add a clear image
Figure 3 : the image of PTEG gear has a spot, it looks like a defect at first, please replace with a clearer image.
Discuss elaborately the variance of theoretical results with experimental results.
The captions need to be more elaborate. Please work on that aspect.
Author Response
Response to Reviewer 2 Comments
Point 1: Line 181: please mention the model of 3D printer used. Also mention if any support material used, also mention the printing angle of the gear with respect to the print bed
Response 1: Thank you for your comment. This paragraph added in Line 172-174 “In the study, BCN 3D Sigma R19 type three-dimensional printer was used to produce plastic gears. No support material was used in the printing process. The printing angle is taken as 0°”.
Point 2: Table 1 : please mention the standards used for measuring the properties (tensile strength etc).
Response 2: Thank you for your comment. The test standarts added in Table 1.
Point 3: Line 193 : Evaluation of 3D printed gears in terms of accuracy and surface finish would be appreciated.
Response 3: Thank you for your comment. This paragraph added in Line 174-176. ‘Since no support material used during the production of the plastic gears, surface finishing processes were not carried out at the end of the production. Accuracy of the produced gears are less then ±0.05 mm.’
Point 4: Figure 1 : the image is blur add a clear image
Response 4: Thank you for your comment. We add a clear image in Line 118.
Point 5: Figure 3 : the image of PTEG gear has a spot, it looks like a defect at first, please replace with a clearer image.
Response 5: Thank you for your comment. We changed the figure of PETG more clearly in Line 180.
Point 6: Discuss elaborately the variance of theoretical results with experimental results.
Response 6: Thank you for your comment. This paragraph added in Line 231-235. ‘When the two methods are compared for the same load and rotational cycles, the experimentally wear values are 4-5% higher on average. Considering the construction times of the tests, the removal of the gears from the device after each test stage and the measurements taken, and the excess of factors affecting the wear on the gears, the use of the theoretical equation is appropriate with this difference.’
Point 7: The captions need to be more elaborate. Please work on that aspect.
Response 7: Thank you for your comment. We checked the manuscript and tried to write the captions in more detail.

Reviewer 3 Report
The manuscript primarily reports wear resistance of plastic spur gears produced by FDM using a theoretical method. Overall, the paper presents a good work. However, at many places, it has too long unnecessary details, while at other places, it lacks some details.
- Literature review is unnecessarily too long. Please make it short highlighting the key points to establish the research gap that is being addressed in the current paper.
- Page#1 (line 41-43): The message is not clear. Please rephrase.
- The manuscript lacks some insights such as (i) what makes PETG better than ABS; is it just the material properties or the printing process as well? (ii) how does the gear performance get affected by change in the printing parameters (iii) how does the FDM printed part performance compare with that of injection molded part of the same materials? Please provide elaboration/ comments on these points.
Author Response
Response to Reviewer 3 Comments
Point 1: Literature review is unnecessarily too long. Please make it short highlighting the key points to establish the research gap that is being addressed in the current paper.
Response 1: Thank you for your comment. The Literature section of the study consists of 3 main parts. The first part is the examination and comparison of the mechanical properties of plastic materials produced with a three-dimensional printer. The second part is the examination of the plastic gears produced by the plastic injection molding method. The third part is the examination of gears produced with three-dimensional printers and finally the purpose of this study, unlike the literature. I agree with you that the literature is long. However, in recent years, journal reviewers want all subjects to be covered in the literature. Short or a small number of literature is the reason for criticism. For this reason, the literature has been comprehensively covered. However, within the scope of your warnings, a few unimportant explanations were deleted and the literature was made short.
Point 2: Page#1 (line 41-43): The message is not clear. Please rephrase.
Response 2: Thank you for your comment. Line 41-43 changed to ‘Compared to other production methods, it is possible to produce products in three-dimensional printers with a low investment cost. Products that have been modeled on three-dimensional printers can be produced in layers without the need for further processing. Unlike other production methods, lighter products can be produced by making the infilling density in the parts. In this system, PLA, ABS, PETG and various thermoplastic polymers can be used in the form of liquid, powder or filament.’
Point 3: The manuscript lacks some insights such as (i) what makes PETG better than ABS; is it just the material properties or the printing process as well? (ii) how does the gear performance get affected by change in the printing parameters (iii) how does the FDM printed part performance compare with that of injection molded part of the same materials? Please provide elaboration/ comments on these points.
Response 3:
- Thank you for your comment. This paragraph added in Line 257-266 ‘In the study, the wear of 3 different polymeric materials produced with a three-dimensional printer at low load and certain rotational repetitions was theoretically investigated. The service life of the gears was determined by continuing the tests until the damage occurred. Since many factors affect the wear of gears, the material properties of gears are emphasized in the study, and the production parameters are made the same for each gear (printing angle, infill density, layer heights, etc.). At the end of the tests, it was observed that the most durable material was PETG. The most important reason for this is that the elongation at break rate is 4-5 times higher than, and this shows that PETG material is more ductile and tough than other materials. PLA and ABS materials showed similar wear behaviors.’
- In this study, only the effect of material properties on wear was investigated theoretically. The explanation on this subject is shown on Line 288-292. ‘In this study, the effect of material properties of three different polymeric materials on wear was investigated theoretically. In future studies, the wear behavior of polymeric gears produced with different three-dimensional printing techniques (SLA, SLS, POM etc.) and printing parameters (infill densities, tooth geometries and raster angles) can be examined comparatively.’
iii. The Nylon-based materials are commonly used in plastic injection method and three-dimensional printers generally. In addition, due to the multiplicity of factors affecting wear in gears, it is necessary to know materials with the same conditions in order to make comparisons. During the literature search, no articles were found about gears produced with the plastic injection method in accordance with the same conditions as this study. The authors wish to explore this topic in their next work.

Round 2
Reviewer 1 Report
Good luck in your work
Author Response
Thank you for your comments.
